# AKBA Promotes Axonal Regeneration via RhoA/Rictor to Repair Damaged Sciatic Nerve

**DOI:** 10.3390/ijms232415903

**Published:** 2022-12-14

**Authors:** Yao Wang, Zongliang Xiong, Chong Zhou, Qiyuan Zhang, Shuang Liu, Sainan Dong, Xiaowen Jiang, Wenhui Yu

**Affiliations:** 1Department of Veterinary Medicine, Northeast Agricultural University, Harbin 150030, China; 2Key Laboratory of the Provincial Education Department of Heilongjiang for Common Animal Disease Prevention and Treatment, Northeast Agricultural University, Harbin 150030, China

**Keywords:** AKBA, sciatic nerve injury, proteomics, axonal regeneration, RhoA/Rictor

## Abstract

The existing studies by our team demonstrated the pro-recovery effect of 3-Acetyl-11-keto-beta-boswellic acid (AKBA) on a sciatic nerve injury. To further investigate the role of AKBA in peripheral nerve injury repair, The TMT quantitative proteomics technique was used to obtain differentially significant proteins in a Sham group, Model group, and AKBA group. After that, three time points (5, 14, and 28 d) and four groups (Sham + AKBA, Sham, Model, and AKBA) were set up, and immunoblotting, immunofluorescence, and cellular assays were applied to investigate the expression of CDC42, Rac1, RhoA, and Rictor in the sciatic nerve at different time points for each group in more depth. The results showed that AKBA enriched the cellular components of the myelin sheath and axon regeneration after a sciatic nerve injury and that AKBA upregulated CDC42 and Rac1 and downregulated RhoA expression 5 d after a sciatic nerve injury, promoting axon regeneration and improving the repair of a sciatic nerve injury in rats. Rictor is regulated by AKBA and upregulated in PC12 cells after AKBA action. Our findings provide a new basis for AKBA treatment of a peripheral nerve injury.

## 1. Introduction

3-Acetyl-11-keto-beta-boswellic acid (AKBA), a pentacyclic triterpene acid found in the gum resin of Boswellia serrata, has sedative–hypnotic effects, and its anti-inflammatory and antioxidant properties protect against neurotoxicity [1,2,3]. AKBA attenuates ischemic neuronal damage in the MCAO model and protects against OGD-induced oxidative stress through the Nrf2/HO-1 signaling pathway [4]. AKBA improves impaired neurological function and reduces cerebral infarction, neuronal cell damage, and apoptosis in a dose-dependent manner, thus fulfilling the neuroprotective potential of AKBA in rats with focal cerebral ischemia [5]. In an experimental model of MeHg-induced ALS, long-term treatment with AKBA at a dose of 100 mg/kg was effective in improving MeHg-induced brain demyelination and suppressing MeHg-induced reduction in MBP expression levels [3]. The above findings suggest that AKBA has a therapeutic effect on different central nervous system injuries. Our research team is dedicated to the therapeutic effects of AKBA on a peripheral nerve injury, and our team members have demonstrated that AKBA promotes the recovery of sciatic nerve injury by increasing the phosphorylation of the ERK signaling pathway and regulating the proliferation of Schwann cells [6,7]. Axon regeneration in a peripheral nerve injury is an important part of injury repair, but the mechanism by which AKBA promotes axon regeneration after a peripheral nerve injury remains to be investigated.

Peripheral nerve damage and axon regeneration are closely associated with Rho proteins. Members of the Rho GTPase family have been shown to regulate many aspects of intracellular actin dynamics and are present in all eukaryotic communities, including yeast and some plants. Ras homolog gene family, member A (RhoA), cell division control protein 42 homolog (CDC42), and Ras-related C3 botulinum toxin substrate 1 (Rac1), as members of the Rho GTPase family, have been implicated in organelle development, cytoskeletal dynamics cell motility, and other common cellular functions. Due to their effects on cell motility and shape, Rho proteins are clear targets for the study of the growth cones formed during axon generation and regeneration in the nervous system. In zebrafish optic nerve regeneration experiments, activation of CDC42 and Rac1 and inactivation of RhoA are required to complete the regeneration process [8]. Knockdown of RhoA enables true axonal regeneration at the lesion site in spinal cord-injured sevengill eels [9]. These studies have demonstrated that inhibition of RhoA expression promotes axonal regeneration in the central nerve. It has also been reported that inhibition of RhoA/ROCK expression in peripheral nerves promotes the neurite growth of sensory neurons in vitro [10]. In vivo, instability of the RhoA pathway promotes functional recovery after a peripheral nerve injury [11]. RhoA activation promotes the assembly of stress fibers (actin-myosin filaments) and adherens patches [12,13]. Rac1 induces membrane puckering and lamellar pseudopod formation [14,15], while CDC42 activation is responsible for the assembly of filamentous pseudopods and actin microspikes [16,17,18,19].

mTOR complex 2 (mTORC2) consists of mTOR (mechanistic target of rapamycin), rapamycin-insensitive MTOR chaperone (Rictor), MLST8, and mammalian stress-activated protein kinase interacting protein 1 (mSIN1). Targeted inhibition of mTORC1 activity by rapamycin could promote functional recovery after a spinal cord injury in rats, by promoting autophagy, anti-inflammation, and neuroprotection [20,21,22]. In contrast, the function of mTORC2 is less well-defined; to date, recent studies have suggested a role for mTORC2 in regulating actin cytoskeleton dynamics [23]. A recent study showed that activation of mTORC2 contributes to the intrinsic axonal growth capacity of adult sensory neurons after injury [24]. Two recent studies showed that downregulation and overexpression of mTORC2 expression affect axonal regeneration after central nerve myelination and a spinal cord injury, respectively [25,26]. Downregulation of mTORC2 expression delays the formation of new myelin sheaths, and its overexpression promotes axonal regeneration after a spinal cord injury.

To further reveal the mechanism of AKBA that promotes the regeneration of damaged axons in peripheral nerves, we obtained differentially expressed proteins 28 days after damage by establishing a sciatic injury model. The analysis of proteomics revealed that the changes of RhoA were different from in previous studies, so, considering the complex process of nerve injury repair, this phenomenon was of concern to us. Using various methods such as histology, immunology, and cell biology, we investigated the regulation of RhoA and its associated Rictor on axon regeneration during the process of an AKBA-promoted sciatic nerve injury repair, using in vitro and in vivo assays. We used immunofluorescence and WB techniques to analyze the mechanism of AKBA on axonal regeneration at the cellular level.

## 2. Results

### 2.1. Bioinformatic Analysis of The Proteomics of Rat Sciatic Nerve Samples

A total of 4936 proteins were identified in the operated lateral sciatic nerve of rats. A hierarchical cluster was used to cluster the differentially expressed proteins in the comparison group, and the data were presented in the form of a heatmap, with the criteria of fold change > 1.2 and *p* value < 0.05 (Student’s *t*-test or one-way ANOVA). We verified the accuracy of the proteomic results by detecting certain proteins in the proteomic results by WB with RT-PCR (Appendix A), the primer sequences were shown in Appendix A. The differentially expressed proteins obtained from the screening can effectively separate the comparison groups and, thus, justify the differentially expressed protein screening (Figure 1A–C). The Cytohubba plug-in of the Cytoscape software was used to screen the key proteins and subnetworks in the differentially expressed proteins of the Model vs. Sham group, the AKBA vs. Sham group, and the AKBA vs. Model group and to construct the reciprocal network graph (Figure 1D–F).

The KEGG pathway was used as a background for the total proteins identified, and Fisher’s exact test (FET) was used to analyze and calculate the significance level of the protein enrichment for each pathway, to identify the significantly affected metabolic and signal transduction pathways. In the Model vs. Sham group, base excision repair, steroid hormone biosynthesis, ABC transporters, amino sugar, nucleotide sugar metabolism, and mineral. The important pathways of base excision repair, steroid hormone biosynthesis, ABC transporters, amino sugar, nucleotide sugar metabolism, and mineral absorption were significantly changed in the AKBA vs. Sham group (Figure 1G). In the AKBA vs. Sham group, significant changes were observed in important pathways such as mineral absorption, galactose metabolism, carbohydrate digestion and absorption, and salivary secretion (Figure 1H). In the AKBA vs. Model group, important pathways such as ribosome, systemic lupus erythematosus, proteoglycans in cancer, proteasome, and the bacterial invasion of epithelial cells underwent significant changes (Figure 1I).

The GO function enrichment analysis of differentially expressed proteins was conducted by Fisher’s exact test. The results showed that in the Model vs. Sham group, there were important biological processes such as the neuron process, neuron process controlling balance, DNA ligation, negative regulation of DNA replication and immune production, molecular functions such as laminin-1 binding, beta-N-acetylhexosaminidase activity, antibiotic transmembrane transporter activity, ferric iron binding, DNA binding, etc. Localized proteins such as synaptonemal structure, synaptonemal complex, basement membrane, and lysosomal lumen underwent significant changes (Figure 1J). In the AKBA vs. Sham group, the cellular response to lipids, axon regeneration, neuron projection regeneration, axon development, peripheral nervous system axon regeneration, nucleic acid binding, ATP-dependent microtubule motor activity, organic cyclic compound binding, DNA topoisomerase binding, telomeric DNA binding, myelin sheath, plasma membrane-bounded cell projection cytoplasm, neuronal cell body, neuron projection, and the somatodendritic compartment, among other localized proteins, underwent significant changes (Figure 1K). In the AKBA vs. Model group, protein–DNA complex assembly, protein–DNA complex subunit organization, DNA packaging, chromatin assembly, nucleosome assembly, and other important biological processes, including structural construction of ribosomes, Rho GDP–disassociation inhibitor binding, nuclear acid binding, GDP–disassociation inhibitor binding, 5′−nuclease activity and other molecular functions, DNA packaging complex, protein–DNA complex, proteasome regulatory particle, base DNA packaging complex, nucleosomes, protein–DNA complex, proteasome regulatory particle, base subcomplex, and cyclosolic ribosomes and other localization proteins, underwent significant changes (Figure 1L).

### 2.2. Toluidine Blue Staining of The Injured Sciatic Nerve in Rats

Toluidine blue staining was performed on the sciatic nerve in each group at different time points to observe the recovery of axons and myelin sheaths after a nerve injury. The sciatic nerve in the Sham + AKBA group and the Sham group had neat and tightly arranged axons and rounded and regular myelin sheath edges at each time point (Figure 2A,B,E,F,I,J). In the Model group and the AKBA group, 5 d after a sciatic nerve injury, the axon arrangement was disorganized and discontinuous, while the myelin sheaths were disintegrated and irregularly shaped (Figure 2C,D). On d 14, the axonal arrangement in the Model group was disorganized, but a tighter arrangement between axons was observed, and the edges of some myelin sheaths were more rounded, while the disintegration of the myelin sheaths was still observed; in the AKBA group, the axonal arrangement was slightly neat and continuous, but the recovery of the myelin sheaths was slightly deficient (Figure 2G,H). On d 28, the recovery of the axons and myelin sheaths was good in the Model group, but misalignment was still observed; the axon arrangement in the AKBA group was slightly disorganized, but the recovery of the myelin sheaths was significantly better than that in the Model group (Figure 2K,L).

### 2.3. Immunofluorescence Detection of Nascent Axons by SCG10

SCG10 can specifically label nascent axons, and we detected the expression of SCG10 by immunofluorescence 5 d, 14 d, and 28 d after a peripheral nerve injury in rats. On the 5th d after sciatic nerve injury, the expression of SCG-10 in the sciatic nerve of SD rats in different treatment groups was significantly lower in the Sham + AKBA group than in the Model and AKBA groups, where the sciatic nerve was not significantly injured, so the immunofluorescence intensity of SCG10 was significantly lower. In the Sham group, there was a significant difference in SCG10 expression compared with the Sham + AKBA group. In the Model group, the fluorescence intensity of SCG10 was significantly higher than that of the Sham + AKBA and Sham groups. In the AKBA group, the immunofluorescence intensity of SCG10 was significantly higher than that of the Sham + AKBA and Sham groups, but the difference between the immunofluorescence intensity of SCG10 and that of the Model group was not significant (Figure 3A–D,M). The fluorescence intensity of SCG10 in the Model group was not significantly different from that in the Sham group. The fluorescence intensity of SCG10 in the AKBA group was significantly stronger than that in the Sham group, and the difference was extremely significant (Figure 3E–H,N). On day 28, the expression of SCG-10 in the four groups was very weak; only the expression of SCG10 in the Model group was significantly different from that in the Sham group, and its expression was lower than that in the Sham group, while the expression of SCG10 in the Sham + AKBA and AKBA groups was not significantly different from that in the Sham group (Figure 3I–L,O).

### 2.4. Protein Expression of The Injured Sciatic Nerve after 5 d of AKBA Intervention

The immunofluorescence figures were quantified by ImageJ software (Figure 4E,F). The mean fluorescence intensity of RhoA was significantly lower in the Sham + AKBA group compared to the Sham group; the mean fluorescence intensity of RhoA was significantly higher in the Model group compared to the Sham group; the mean fluorescence intensity of RhoA was significantly higher in the AKBA group compared to the Sham group; and the mean fluorescence intensity of RhoA was significantly higher in the AKBA group compared to the Sham group. There was no significant difference in the mean fluorescence intensity of RhoA in the AKBA group compared to the Model group; the mean fluorescence intensity of Rictor was significantly higher in the Sham + AKBA group compared to the Sham group; the mean fluorescence intensity of Rictor was significantly higher in the Model group compared to the Sham group; and the mean fluorescence intensity of Rictor was significantly higher in the Model group compared to the Sham group. The mean fluorescence intensity of Rictor was significantly higher in the AKBA group compared to the Sham group (Figure 4A–D). The immunoblotting results of RhoA and Rictor were consistent with the immunofluorescence results, and the Sham + AKBA group had a significantly lower CDC42. The expression level of CDC42 in the Sham + AKBA group was significantly lower than that in the Sham group. The expression level of CDC42 in both the Model and AKBA groups was significantly higher than that in the Sham group. There was no significant difference in the expression level of CDC42 in the Model and AKBA groups. Rac1 expression was significantly downregulated in the Sham + AKBA group compared to the Sham group, with no significant change in the Model group, and it was significantly upregulated in the AKBA group. Rac1 expression was significantly lower in the Model group than in the AKBA group (Figure 4G–K).

### 2.5. Protein Expression of The Injured Sciatic Nerve 14 d after AKBA Intervention

The immunofluorescence figures were quantified by ImageJ software (Figure 5E,F). The mean fluorescence intensity of RhoA was significantly lower in the Sham + AKBA group compared to the Sham group; the mean fluorescence intensity of RhoA was significantly lower in the Model group compared to the Sham group; and the mean fluorescence intensity of RhoA was significantly lower in the AKBA group compared to the Sham group. The mean fluorescence intensity of RhoA in the AKBA group was significantly higher than that in the Model group compared to the Sham group. The mean fluorescence intensity of Rictor in the Sham + AKBA group was significantly lower than that in the Sham group; and the mean fluorescence intensity of Rictor in the Model group compared to the Sham group was significantly. The mean fluorescence intensity of Rictor in the AKBA group was significantly lower than that in the Sham group; and the mean fluorescence intensity of Rictor in the Model group was significantly higher than that in the Sham group (Figure 5A–D). The immunoblotting results of RhoA and Rictor were consistent with the immunofluorescence results. Both CDC42 and Rac1 were significantly upregulated in the AKBA group compared with the Sham group; CDC42 was significantly upregulated, and Rac1 was not significantly changed in the Model group; and CDC42 was not significantly changed, while Rac1 was significantly upregulated, in the Sham + AKBA group (Figure 5G–K).

### 2.6. Protein Expression of The Injured Sciatic Nerve after 28 d of AKBA Intervention

The immunofluorescence figures were quantified by ImageJ software. There was no significant difference in the mean fluorescence intensity of RhoA between the Sham + AKBA group compared with the Sham group; the mean fluorescence intensity of RhoA was significantly lower in the Model group compared with the Sham group; the mean fluorescence intensity of RhoA in the AKBA group, when compared with the Sham group, was significantly lower in the Sham group; and there was no significant difference in the mean fluorescence intensity of RhoA between the Sham + AKBA group compared with the Sham group. The mean fluorescence intensity of RhoA in the AKBA group was significantly higher than that in the Sham group compared with the Model group. On d 28 after a sciatic nerve injury, the expression of Rictor was not significantly different between the groups (Figure 6A–D). The immunoblotting results of RhoA and Rictor were consistent with the immunofluorescence results. Compared with the Sham group, there was no significant change in CDC42 in the Sham + AKBA and Model groups; CDC42 was significantly upregulated in the AKBA group; and Rac1 was significantly downregulated in the Sham + AKBA, Model, and AKBA groups (Figure 6G–K).

### 2.7. Changes in Protein Expression after AKBA Action on PC12 Cells

The cells were divided into three groups according to the treatment: Ctrl^+^ group (DMEM + 10% FBS), Ctrl^−^ group (DMEM), and AKBA group (AKBA 5 μM) (Appendix A). The expression of RhoA, Rictor, SCG10, and βIII-Tubulin was detected by immunofluorescence. The fluorescence intensity of RhoA in the AKBA group was lower than that in the Ctrl^+^ and Ctrl^−^ groups (Figure 7A–C,M). The fluorescence intensity of SCG10 was higher in the AKBA group than in the Ctrl^+^ and Ctrl^−^ groups (Figure 7G–I,O). PC12 cells treated with AKBA improved the extension of neurites, and the length of new neurite protrusions in the AKBA group was closer to that in the Ctrl^+^ group than in the Ctrl^−^ group (Figure 7J–L,P). After collecting the cells for protein extraction, the expression of RhoA, CDC42, Rac1, and Rictor was detected by immunoblotting. Using the Ctrl^−^ group as a control, CDC42 was significantly upregulated in both the Ctrl^+^ and AKBA groups; Rac1 and Rictor were significantly upregulated, while RhoA was significantly downregulated, in the AKBA group; and Rac1 was significantly downregulated, while RhoA and Rictor were not significantly changed, in the Ctrl^+^ group (Figure 7Q–U).

## 3. Discussion

### 3.1. Effect of AKBA Intervention on the Bioinformatics of Late SNI

Ubiquitin C-terminal hydrolase L1 (Uchl1) as well as neurofilament light (Nefl) are involved in neuromuscular biological processes related to axonal transport, axonal regeneration, and axonal target recognition. The enrichment between the Model and Sham groups was significant mainly for biological processes, of which the GO:0050905 neuromuscular process was the most enriched. The biological processes of the neuromuscular process were significantly enriched on the 28th d after a sciatic nerve injury, but were no longer significantly enriched in the AKBA and Sham comparison groups, indicating that the injured nerve in the Model group was associated with the functional reconstruction of the original effector organ that was underway in the Model group. The enrichment between the AKBA and Sham groups was mainly in the cellular fractions, of which the GO:0043209 myelin sheath (CC) and GO:0031103 axon regeneration (BP) were the most enriched, and the effect of AKBA on the recovery of the myelin sheath after nerve injury has been confirmed [7]. The effect of AKBA on the recovery of the myelin sheath after a nerve injury has been demonstrated, and AKBA can regulate the repair of a myelin injury in the sciatic nerve by promoting the differentiation of the Schwann cells. The proteins enriched in the biological process of axonal regeneration in the AKBA group and the Sham comparison group had differences from the proteins involved in the Model group and the Sham comparison group, and apolipoprotein E (ApoE) and microtubule-associated protein 1B (Map1b) are involved in the biological process of axonal regeneration. Previous studies have shown that ApoE expression promotes the repair of nerve injury [18]. The absence of Map1b reduced the amount of acetylated microtubule proteins and caused swelling of the neuronal axons of DRG as well as the protrusion and contraction of the filamentous pseudopod-like spines [27]. The enrichment between the AKBA and Model groups was mainly in the cellular fractions, with the GO:0000786 nucleosome (CC) being the most significantly enriched, while the protein–DNA combinations were similarly enriched in this comparison group. This indicates that 28 d after a sciatic nerve injury, AKBA promotes repair by regulating the synthesis of protein and DNA. The above results indicated that the recovery of axons and myelin sheaths was similar between the AKBA and Model groups on the 28th d after a sciatic nerve injury. Combined with the recovery course after a peripheral nerve injury, on d 28, the nerve after a sciatic nerve injury had successfully linked with the original effector organ and started the subsequent repair, which explains the absence of significant enrichment of axon- and myelin-related GO entries in the comparison groups of the AKBA and Model groups.

The most significant enrichment between the AKBA group and the Model group was in the ribosome. The expression of RPL4, RPL23a, RPL35, RPS11, RPL34, RPL13a, RPL10, RPS6, RPS8, RPS24, and RPL6, which are involved in the ribosomal pathway proteins, were significantly upregulated, while the expression of RPL12, MRPL1, and RPS25 was significantly downregulated. It has been shown that both wild-type mice and spinal muscular atrophy (SMA) mice contain ribosomes in their motor axons, and SMA mice have a 27% reduction in their axonal ribosomes compared to wild-type mice [28]. There were increased phosphorylated S6 ribosomal protein in the axons after a sciatic nerve injury [29]. Interestingly, L10a and ribosomal 5.8S RNA are required in the sensory neuron axons to translate local mRNA [30]. Therefore, it can be hypothesized that AKBA regulates the protein synthesis of axonal ribosomes through these proteins, thus improving the repair of a sciatic nerve injury.

After a nerve injury, increased TGF-β promotes the production of inhibitory proteins by astrocytes, to hinder neuropil growth [31]. RhoA was differentially significant in the TGF-β pathway that was significantly enriched in the AKBA group vs. the Model group. RhoA was significantly upregulated between the AKBA group and the Model group, but RhoA was not significantly altered in the AKBA group vs. the Sham group, or the Model group vs. the Sham group. It has been shown that RhoA, as one of the main proteins regulating the actin cytoskeleton, and the downregulation of RhoA can promote the axonal regeneration and differentiation of Schwann cells after a nerve injury [9,32,33]. The repair of a sciatic nerve injury is a complex and slow process, and the process is divided into three phases: pre, mid, and post. In the first stage, as most studies have shown, the downregulation of RhoA expression promotes axonal regeneration, and when RhoA is overexpressed, the extension of nerve protrusions is significantly inhibited [34]. The reports on RhoA expression in the middle and late stages of a nerve injury repair are less frequent. However, based on the repair process of a nerve injury, it can be speculated that the downregulation of RhoA expression is needed to promote the regeneration of axons and the extension of growth cones in the early stage, to facilitate the reconstruction of the functional linkage between the nerve and the original effector organ; once the reconstruction of the functional linkage is completed, the removal of redundant axons and the removal of faulty functional linkage are needed. At this point, RhoA expression upregulation can inhibit the regeneration of the remaining axons and the extension of the growth cone, facilitating the removal of redundant axons after neural link reconstruction. It has been shown that the interaction between RhoA and Rictor can regulate the actin cytoskeleton [35,36]. It has also been shown that enhancement of the mTORC2 pathway can improve functional recovery in spinal cord injured rats [37]. To test this speculation, the changes in mTORC2 (Rictor) during the SNI recovery process were examined, and the results showed that Rictor expression in the AKBA group was suppressed by AKBA in the early SNI phase and upregulated by AKBA in the middle SNI phase. This is a very interesting difference, which indicates that Rictor expression in the SNI process does not follow a regular pattern but has different expressions at different times and regulates different functions. We will unravel its mystery in a subsequent study.

### 3.2. Effect of AKBA Intervention on Regeneration of Injured Sciatic Nerve

In the experiments with the Sham + AKBA group, where the sciatic nerve was not damaged, no damaging effect of AKBA was observed, indirectly confirming the safety of AKBA as a therapeutic agent for a peripheral nerve injury. Based on the immunofluorescence results of SCG10, it is known that the nerve in the Sham group still caused slight damage to the nerve during modeling using surgical operations such as the simple exposure of the nerve, and this result suggests that we should optimize the surgical operations during modeling to reduce surgical injury. On d 5, high SCG10 expression was observed in the sciatic nerve of the Sham group, but SCG10 expression dropped rapidly on d 14 to a level similar to that on d 28. This indicates that the damage caused to the Sham group at the time of modeling was healed within 14 d by the body’s ability to repair itself. Therefore, if the samples were taken for follow-up experiments after a short-term pharmacological intervention, a slight injury in the Sham group would still interfere with the experimental results. On d 14, the expression level of SCG10 was not significantly different from that of the Sham group, and, on d 28, the expression level of SCG10 was extremely low, as the number of new axons had been significantly reduced at this time. The results of the Model group showed that two weeks after a sciatic nerve injury, the number of new axons had decreased significantly compared with on d 5, and the decrease in the number of new axons could be due to the inflammatory environment after the nerve injury hindering the regeneration of axons or the normal decrease in the number of new axons with the gradual recovery of function. After intervention with AKBA, in the whole healing process of a sciatic nerve injury, axonal regeneration is only the first step of the whole process. The regeneration of axons is the basis of a peripheral nerve injury, and the new axons are responsible for the reconstruction of the functional link between the injured nerve and the original effector organ, since the subsequent myelin formation will take place only after the re-linking. From the changes in the number of new axons, we can learn that axon regeneration was still active on the 5th day after the sciatic nerve received a crush injury, and, on the 14th day, the number of new axons decreased significantly compared with the 5th day. Immunofluorescence of SCG10 corroborated the results of GO and KEGG in proteomics and explained the absence of axon- and myelin-related enrichment between the AKBA and Model groups.

### 3.3. Effect of AKBA Intervention on Axon-Related Protein Expression in Injured Sciatic Nerve

According to the time points of myelin recovery after nerve injury [38], three time points of observation were set up in this experiment, representing the early (5 d), middle (14 d), and late (28 d) periods after nerve injury. During the repair process after a nerve injury, there are contrasting biological processes in the early and late stages. In the early stage, a large number of new axons need to be generated to reconstruct the original function of the interneurons, and, during this process, the increased expression of Rac1 and CDC42 promotes the regeneration of axons. The results of this study showed that the expression of CDC42 was not affected by AKBA in the early stage, and its expression level was not significantly different from that of the Model group. However, in the middle stage, the expression of CDC42 was significantly upregulated by AKBA, the regulation of Rac1 by AKBA continued throughout the damage repair process, and the expression of Rac1 in the AKBA group was significantly higher than that in the Model group at all time points. It has been shown that the expression of the dominant inactivating mutants of Rac1 promote neurite growth in dorsal root ganglion (DRG) cells [10]. In contrast, constitutively active Rac1 increased the proportion of collapsed growth cones in DRG neurons [29,38]. Combined with the process of nerve injury repair, AKBA may promote the expression of the dominant inactivating mutants of Rac1 for axon regeneration in the pre-SNI period and collapse and atrophy the free growth cones by increasing the expression of constitutively active Rac1 in the later period, which, in turn, removes the free axons and trims the repaired intact nerve fibers. An elaboration of this interesting discrepancy is required by our subsequent studies.

Proteomics showed that the expression of RhoA was significantly different from the results of the existing studies. The expression of RhoA was suppressed in the pre-injury period and promoted axonal regeneration and growth cone migration. In our results, we found that the expression of RhoA was not significantly different between the Model group and the AKBA group at the early stage of a nerve injury, which may be influenced by the insignificant effect of AKBA on its expression or because the regenerative environment formed by the disintegrated myelin sheaths and myelin fragments hindered the extension of axons at the early stage of an injury. At the middle stage of a nerve injury, AKBA significantly downregulated RhoA expression and promoted axon regeneration. Rictor overexpression may also contribute to this process during this period, as Rictor overexpression plays a role in the anti-inflammatory response and drives macrophage polarization toward the M2 phenotype, so changes in Rictor may be related to the inflammatory response during nerve repair as well as to macrophages. The expression of Rictor in the pre-injury period was significantly higher in the Model group than in the AKBA group, which is different from the existing studies. However, in the mid-injury period, the expression level of Rictor was significantly higher in the Model group in response to AKBA. Overexpression of RhoA induces neurosynaptic contraction and growth cone collapse. Our results also demonstrate an upregulation of RhoA in the late stages of injury as well as a substantial decrease in the number of nascent axons. This may be due to the fact that at the late stage of a nerve injury, the functional linkage of axons to the target organs has been completed, while the redundant axons that have not yet been successfully linked need to be removed. SCG10 staining also indicates that at the late stage, the number of newborn axons is close to 0. Therefore, at the late stage of injury, AKBA repairs the axons by both the atrophy of growth cones and the phagocytosis of macrophages through Rictor and RhoA. The final process of injured nerve repair is completed. In cellular experiments, the neuropil of PC12 cells was significantly superior to the Ctrl-group in the presence of AKBA, and the changes in RhoA downregulation and Rictor upregulation were also consistent with the changes in the injured nerve.

These results showed that AKBA showed a certain time-dependent effect on the repair of a sciatic nerve injury, and the regulation of axonal-regeneration-related proteins such as CDC42 and RhoA by AKBA was not obvious in the early stage of a nerve injury. The expression of CDC42 and RhoA was significantly upregulated or downregulated with the increase in drug-administration time. The expressions of CDC42 and RhoA in the later stages of a nerve injury were different from or even opposite of those in the earlier stages, and AKBA significantly upregulated the expression of Rictor in the middle stage of injury to maintain neuronal survival through an anti-inflammatory response.

## 4. Methods and Materials

### 4.1. Establishment of a Model of Sciatic Nerve Injury in Rats

Sprague Dawley rats were purchased from Liaoning Changsheng (Liaoning, China), and the rats were first acclimatized for one week. After the acclimatization, the right sciatic nerve was surgically exposed to establish a sciatic nerve injury model [39]. Sham group SD rats were only exposed to the sciatic nerve without clamping treatment of the injured nerve. The rats were fed and watered at any time during the feeding process. The experimental animals were handled following the regulations related to the animal welfare of Northeastern Agricultural University (NEAUEC2019 05 08).

### 4.2. Experimental Grouping and Drug Administration in Rats

The modeled SD rats were randomly divided into the Model group and the AKBA dosing group, with 25 rats in each group. The Sham group was composed of 25 SD rats with only the sciatic nerve exposed, and the other 15 SD rats without any surgical treatment were taken as the blank dosing group. The rats in the AKBA group and the rats in the blank group were administered by gavage (10 mg/kg) on days 1, 3, 5, 7, …, 27 after modeling, and the model control group and the Sham group were gavaged with equal amounts of saline at the same time, to ensure that the administration time of each gavage was as consistent as possible [40]. The rats in the model control group and Sham group were administered equal amounts of saline at the same time.

### 4.3. Proteomic Analysis of Rat Sciatic Nerve Samples

TMT quantitative proteomics analysis was performed by Applied Protein Technology (Apt, Shanghai, China). The proteomic results were analyzed bioinformatically utilizing Chi’s method [41]. Briefly, the obtained differentially expressed proteins were analyzed by clustering, enrichment analysis of Kyoto Encyclopedia of Genes and Genomes (KEGG) and Gene Ontology (GO), and protein interactions by Strings.

### 4.4. Western Blot Assay of Rat Sciatic Nerve Samples

Sciatic nerve proteins were extracted according to the whole protein extraction kit (Strong) (Beijing, Solarbio). Protein concentrations were calculated using the BCA protein assay. Immunoblotting methods were performed according to Jiang’s experimental method [6] The amount of sciatic nerve tissue protein sampled was 5–20 μg/well; after electrophoresis, the proteins were transferred to polyvinylidene fluoride (PVDF) membranes (BioSharp, China). Waiting for the incubation of primary and secondary antibodies to finish, exposure was performed. Image information was collected, and the results were quantified by ImageJ software.

### 4.5. Toluidine Blue Staining of Rat Sciatic Nerve Samples

The tissue sections were first routinely dewaxed to water and then put into toluidine blue solution for 30 min, followed by a slight wash into a glacial acetic acid solution for differentiation until the nuclei and granules were clear. After that, the sections were slightly washed with water and dried with cold air. Then, xylene was transparent, and neutral gum was used to seal it. Finally, microscopic observation was performed to obtain images.

### 4.6. Immunofluorescence Detection of Rat Sciatic Nerve Samples

Immunofluorescence according to Catenaccio’s method [42]. Briefly, sections were dewaxed to water and placed in an antigen repair solution for antigen repair. The sections were slightly shaken dry and then closed by drawing circles around the tissue with a histochemical pen (to prevent antibody runoff); drops of BSA-blocking solution were added to the circles. After removing the blocking solution, the primary antibody was added and incubated overnight at 4 °C. The secondary antibody was added and incubated at room temperature with protection from light. Autofluorescence quencher was added to the circles and shook dry slightly, and the slices were sealed with anti-fluorescence quenching sealer. Finally, images were observed and acquired using a fluorescence microscope. Average fluorescence intensity analysis was performed using ImageJ software.

### 4.7. Cytotoxicity Assay of PC12 Cells

Cells were incubated at 37 °C in a 5% CO_2_ incubator. PC12 cells were grown against the wall, and cell passages were performed when the cell density in the field of view reached 80% or more under the microscope. The suitable concentration of AKBA acting on PC12 cells was detected according to the CCK-8 Cell Proliferation-Toxicity Assay Kit. Briefly, cell suspensions (100 μL/well) were inoculated in 96-well plates and incubated for 12 h or 24 h afterward. Then, 10 μL CCK-8 solution was added to each well, which was incubate in a cell incubator for 30 min, and the absorbance value at 450 nm was measured with an enzyme specimen.

### 4.8. Effect of AKBA on PC12 Cells by Immunofluorescence Detection and Immunoblotting

PC12 cell suspensions were inoculated in 48-well plates, given in DMEM, AKBA, and DMEM + 10% FBS and incubated for 24 h. Then, the culture medium was discarded, and paraformaldehyde was added, fixed for 30 min, and closed overnight by adding blocking serum, followed by incubation of primary and secondary antibodies. Incubation of the fluorescent secondary antibody requires a light-proof operation. DAPI is used to re-stain the cell nuclei and incubate at room temperature away from light. An autofluorescence quencher was added to the circles and slightly shaken dry, and the anti-fluorescence quenching sealer was used to seal the slices. Finally, images were observed and acquired using a fluorescence microscope. Average fluorescence intensity analysis was performed using ImageJ software.

Protein extraction of PC12 cells was performed according to the Whole Protein Extraction Kit (Strong) (Beijing, Solarbio). Briefly, the medium was discarded and washed twice with cold PBS, the PBS was discarded, then the calculated cell lysate was added; the cells were scraped off with a cell scraper on ice, the scraped cell lysate was transferred into an EP tube and was lysed upside down for 20–30 min. after lysis; the cell lysate was centrifuged at 4 °C at 12,000× *g* for 30 min, and the supernatant was transferred into a new EP tube. Protein quantification was performed and then the PC12 cells were denatured for protein experiments (Table 1).

### 4.9. Statistical Analysis

All data were statistically analyzed using GraphPad Prism Version 8.3 to analyze the experimental data, which were plotted by GraphPad Prism Version 8.3 software. All values were expressed in the form of means ± SD. One-way ANOVA analysis followed by Tukey’s post hoc test was used to perform multiple comparisons. Minimum significance levels were set at * *p* < 0.05, *** p* < 0.01, *** *p* < 0.001, and ***** p* < 0.0001.

## 5. Conclusions

AKBA enriches the cellular components of the myelin sheaths and axonal regeneration after a sciatic nerve injury, and AKBA improves the repair of a sciatic nerve injury in rats by upregulating CDC42 and Rac1 and downregulating RhoA expression at different time points. Rictor is regulated by AKBA during sciatic nerve injury repair and is upregulated in PC12 cells after AKBA action.

## Figures and Tables

**Figure 1 ijms-23-15903-f001:**
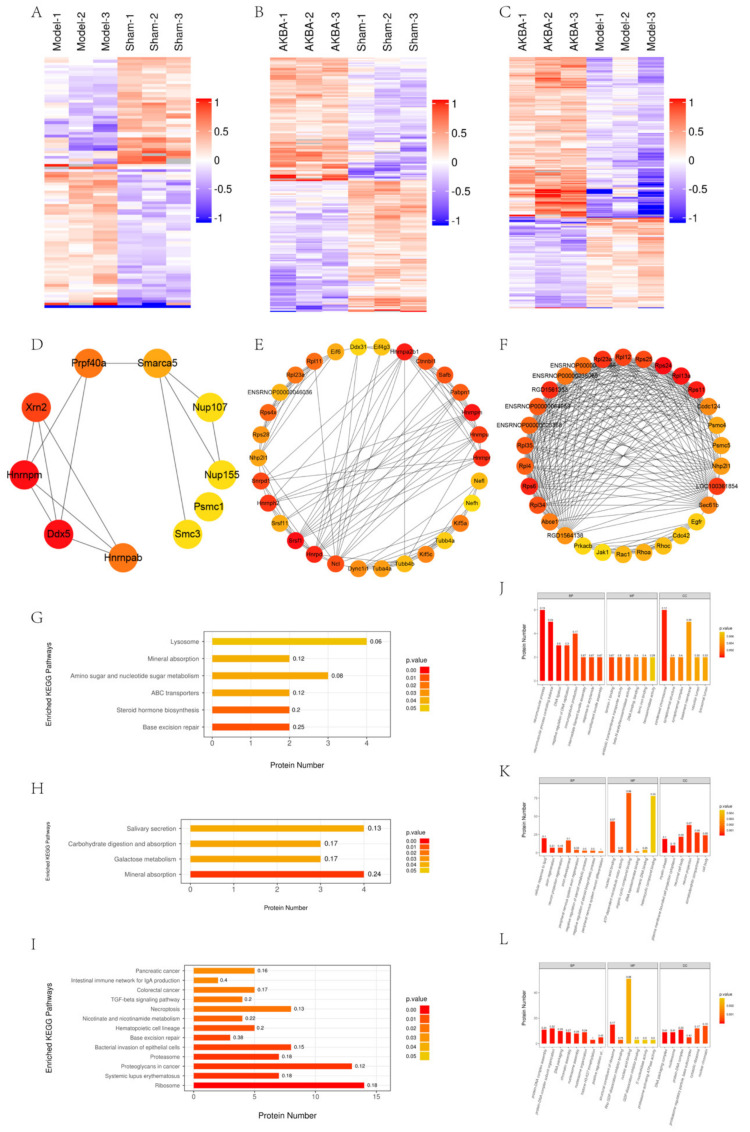
Bioinformatics analysis of proteomics. (**A**–**C**): Each row in the tree heatmap represents one protein (i.e., the vertical coordinate is the significantly differentially expressed protein, each column represents a group of samples, red represents significantly upregulated proteins, blue represents significantly downregulated proteins, and the gray part represents no quantitative information about proteins. ABC is the Model vs. Sham group, AKBA vs. Sham group, and AKBA vs. Model group, respectively. The number of key proteins in Model vs. Sham group (**D**) is set to 10, and the number of key proteins in AKBA vs. Sham group (**E**) and AKBA vs. Model group (**F**) is set to 30. (**G**–**I**): The vertical coordinate indicates the significantly enriched KEGG pathways; the horizontal coordinate indicates the number of differentially expressed proteins in each KEGG pathway; the color of the bar indicates the significance of the enriched KEGG pathways. The *p* value is calculated based on Fisher’s exact test, and the color gradient represents the *p* value: the closer to red means the smaller the *p*-value, and the higher the significance level of the enriched KEGG pathway; the label above the bar shows the enrichment factor (richFactor ≤ 1), and the enrichment factor indicates the ratio of the number of differentially expressed proteins involved in a KEGG pathway to the number of proteins involved in the pathway among all the identified proteins. GHI is (**J**–**L**): the horizontal coordinates indicate the enriched GO functional categories, which are divided into biological process (BP), molecular function (MF), and cellular component (CC). JKL is the Model vs. Sham group, AKBA vs. Sham group, and AKBA vs. Model group, respectively.

**Figure 2 ijms-23-15903-f002:**
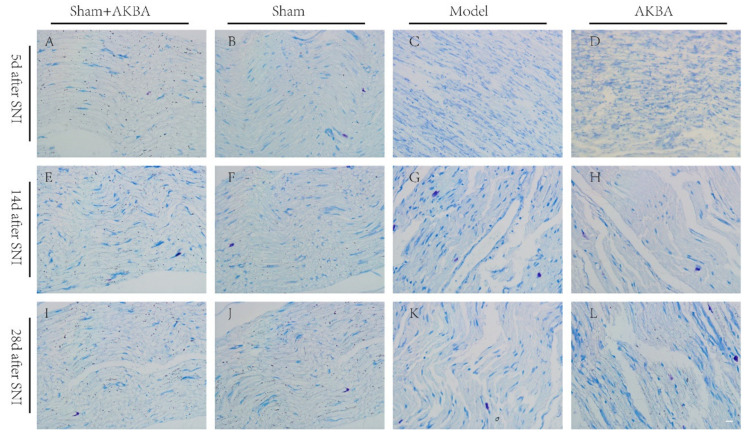
Toluidine blue staining of the sciatic nerve in each group at different time points. (**A**–**D**) Toluidine blue staining of the sciatic nerve in each group 5 d after injury. (**E**–**H**) Toluidine blue staining of the sciatic nerve in each group 14 d after injury. (**I**–**L**) Toluidine blue staining of the sciatic nerve in each group 28 d after injury. The scale bar of all images is 20 μm.

**Figure 3 ijms-23-15903-f003:**
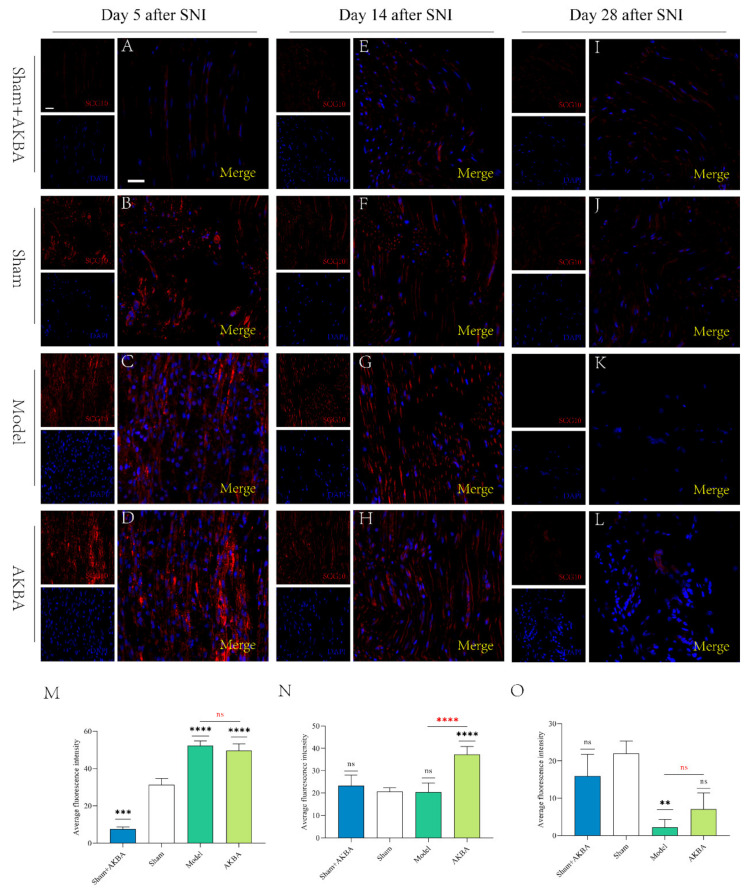
SCG10 staining at various time points after sciatic nerve injury. (**A**–**D**) Immunofluorescence of SCG10 d 5 after sciatic nerve injury. (**E**–**H**) Immunofluorescence of SCG10 d 14 after sciatic nerve injury. (**I**–**L**) Immunofluorescence of SCG10 d 28 after sciatic nerve injury. (**M**–**O**) Quantification of mean fluorescence of SCG10 immunofluorescence 5 d, 14 d, and 28 d after sciatic nerve injury. Black * is the comparison with the Sham group, and red * is the comparison of the Model group with the AKBA group. ** *p* < 0.01, *** *p* < 0.001, **** *p* < 0.0001; ns: no significant difference. The subsequent images are labeled similarly. The scale bar of all images is 20 μm.

**Figure 4 ijms-23-15903-f004:**
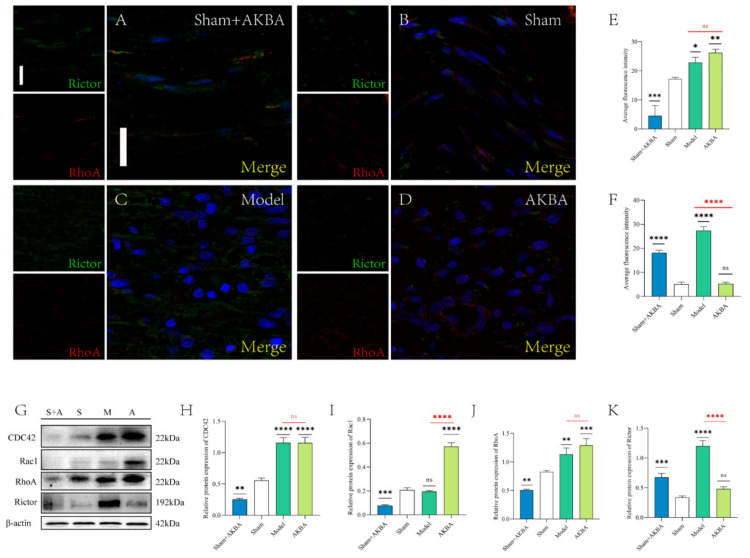
Protein changes in the injured sciatic nerve after 5 d of AKBA intervention. (**A**–**D**) Immunofluorescence staining plots of RhoA and Rictor for each group. The scale bar of the images is 20 μm. (**E**–**F**) Quantification of the mean fluorescence intensity of RhoA and Rictor. (**G**) Immunoblot plots of RhoA, CDC42, Rac1, and Rictor for each group. (**H**–**K**) Quantification of protein expression of CDC42, Rac1, RhoA, and Rictor, respectively, in order. Levels were quantified. Black * is the comparison with the Sham group, and red * is the comparison of the Model group with the AKBA group. * *p* < 0.05, ** *p* < 0.01, *** *p* < 0.001, **** *p* < 0.0001; ns: no significant difference.

**Figure 5 ijms-23-15903-f005:**
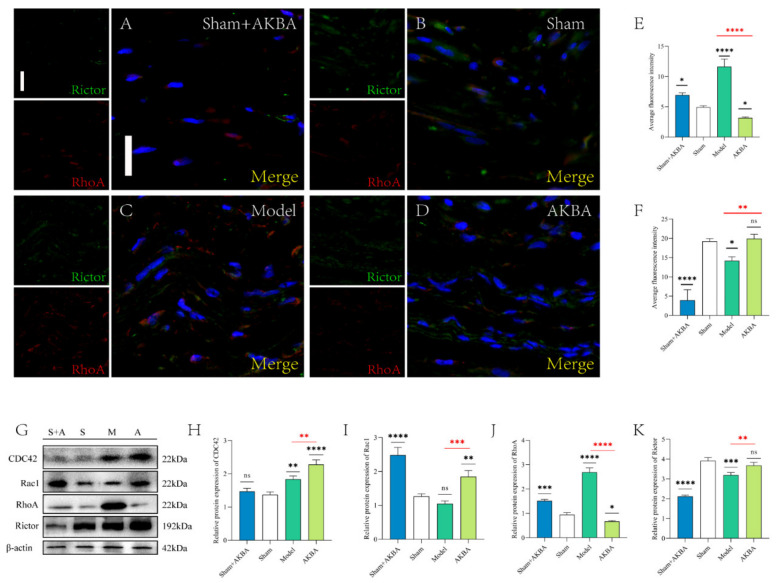
Protein changes in the injured sciatic nerve after 14 d of AKBA intervention. (**A**–**D**) Immunofluorescence staining plots of RhoA and Rictor in each group. The scale bar of the images is 20 μm. (**E**–**F**) Quantification of the mean fluorescence intensity of RhoA and Rictor. (**G**) Immunoblot plots of RhoA, CDC42, Rac1, and Rictor in each group. (**H**–**K**) Quantification of protein expression of CDC42, Rac1, RhoA, and Rictor, respectively, in order. Levels were quantified. Black * is the comparison with the Sham group, and red * is the comparison of the Model group with the AKBA group. * *p* < 0.05, ** *p* < 0.01, *** *p* < 0.001, **** *p* < 0.0001; ns: no significant difference.

**Figure 6 ijms-23-15903-f006:**
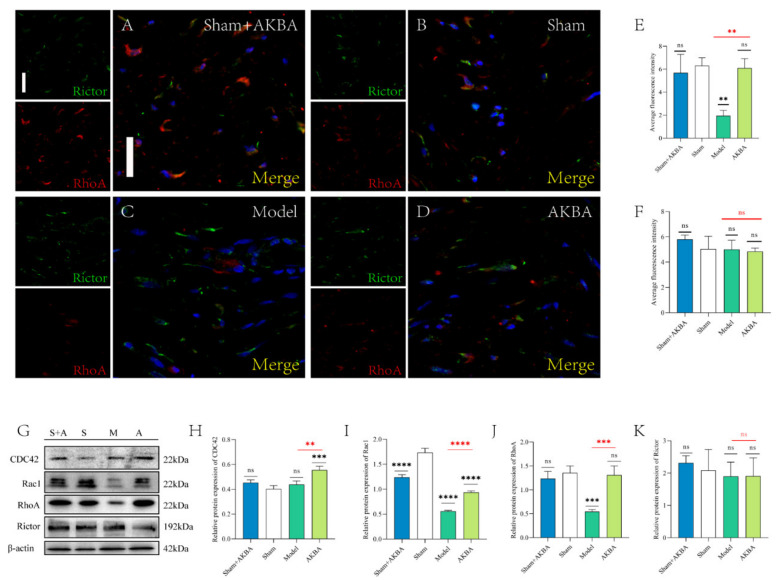
Protein changes in the injured sciatic nerve after 28 d of AKBA intervention. (**A**–**D**) Immunofluorescence staining plots of RhoA and Rictor for each group. The scale bar is 20 μm. (**E**–**F**) Quantification of the mean fluorescence intensity of RhoA and Rictor. (**G**) Immunoblot plots of RhoA, CDC42, Rac1, and Rictor for each group. (**H**–**K**) Quantification of the protein expression of CDC42, Rac1, RhoA, and Rictor, respectively, in order. Levels were quantified. Black * is the comparison with the Sham group, and red * is the comparison of the Model group with the AKBA group. ** *p* < 0.01, *** *p* < 0.001, **** *p* < 0.0001; ns: no significant difference.

**Figure 7 ijms-23-15903-f007:**
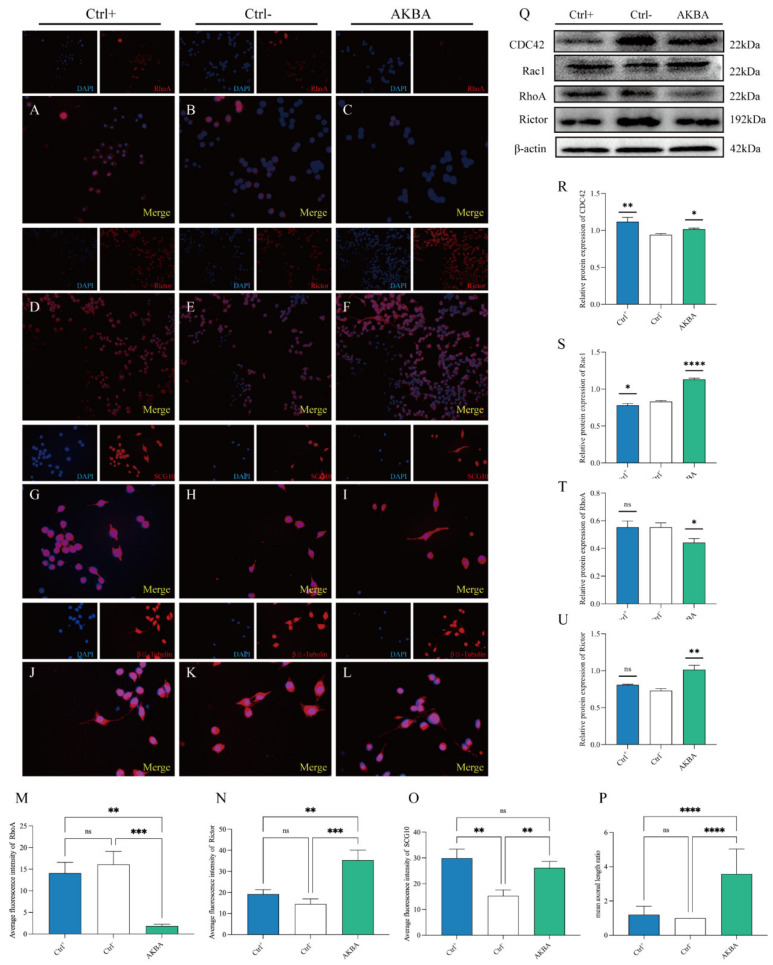
Improved growth of neuronal synapses in PC12 cells after AKBA treatment. (**A**–**C**) RhoA immunofluorescence staining of different groups of PC12 cells. (**D**–**F**) Rictor immunofluorescence staining of different groups of PC12 cells. (**G**–**I**) SCG-10 fluorescence staining of different groups of PC12 cells. (**J**–**L**) βIII-Tubulin fluorescence staining of different groups of PC12 cells. (**M**) Quantification of the mean fluorescence intensity of RhoA immunofluorescence. (**N**) Quantification of the mean fluorescence intensity of Rictor immunofluorescence. (**O**) Quantification of the mean fluorescence intensity of SCG10 immunofluorescence. (**P**) Mean length of neuropil, *n* ≥ 5. (**Q**) Effect of AKBA on protein expression of RhoA, CDC42, Rac1, and Rictor after action on PC12 cells. (**R**–**U**) Quantification of the grayscale of the immunoblotted bands. * *p* < 0.05, ** *p* < 0.01, *** *p* < 0.001, **** *p* < 0.0001; ns: no significant difference.

**Table 1 ijms-23-15903-t001:** Antibodies used in the test.

Antibody	Application	Cat. No	Brands
RhoA	WB, IF	BM4479	Boster, China
CDC42	WB	bs-3555R	Bioss, China
Rac1	WB	A7720	Abclonal, China
ApoE	WB	WL03172	Wanlei, China
Rictor	WB	WL04494	Wanlei, China
Rictor	IF	MB0063	ABmart, China
SCG10	IF	A04729–2	Boster, China
βIII-Tubulin	IF	A17913	Abclonal, China
P16 ARC	WB	AF1069	Beyotime, China

## Data Availability

Not applicable.

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
