# Peer review of "AKBA Promotes Axonal Regeneration via RhoA/Rictor to Repair Damaged Sciatic Nerve"

_ijms, 2022, doi:10.3390/ijms232415903_

Round 1

Reviewer 1 Report

Xiong et al reported an interesting manuscript regards the role of AKBA in peripheral nerve injury repair. There are some suggestions that should be taken into consideration further.

1. Rationale for why only three-time points (5, 14, and 28 d) were used? Since AKBA revealed some time dependence on the repair of sciatic nerve injury, and the regulation of axonal regeneration-related proteins such as CDC42 and RhoA by AKBA is not obvious in the early stage of nerve injury, Long-term effects as 2 months need to be additionally performed. 

2. Text in Fig. 1. J-L is hard to read

3. Changes in protein expression between days 5, 14, and 28 were not well described and discussed

4. Since all Immunoblot plots were not really clear, please provide the uncropped image in the supplement

5. Immunofluorescence staining images need to improve their quality, some dots seemed to be not targeted staining...

Reviewer 2 Report

Dear author, 

I hope you are well and safe while I write this. The present research aimed to assess AKBA's function in mending nerve damage at the periphery. The author used the  TMT quantitative proteomics to identify considerably different proteins across the Sham, Model, and AKBA groups. The author concludes that KBA enhances the healing of sciatic nerve damage in rats by upregulating CDC42 and Rac1 expression and decreasing RhoA expression at different periods. AKBA enriches the cellular components of the myelin sheath and axonal regeneration following sciatic nerve injury. AKBA controls Rictor expression during sciatic nerve regeneration and AKBA-induced Rictor upregulation in PC12 cells.

The author's findings had a considerable outcome of nerve damage at the periphery. However, I recommend the author revise the following:

1-     Revise the article for abbreviation showed stated in full words for the first time in the manuscript.

2-     Revise the article for missing spaces, especially before references.

3-     In line 102, please replace SD with Sprague Dawley.

4-     The author should provide the ethical approval number of the ethical committee in the methods.

5-     In the statistical analysis, most of the data analyzed are considered non-parametric. Therefore the author should state how the data was validated for the utilization of a parametric test (ANOVA). 

6-     The figures’ signs of significance are not stated in the captions. The authors are required to provide to make the figures self-explanatory.

7-     Lines 412 and 413 are not clear and should be revised.

8-     In the discussion section, the author should state a brief sentence about the meaning of enrichment in lines 415 and 416.

Yours

Round 2

Reviewer 1 Report

Thank you for addressing and clarifying my concerns.  I have no further comment. The manuscript can now be acceptable!